# Mitochondrial Oxidative Stress—A Causative Factor and Therapeutic Target in Many Diseases

**DOI:** 10.3390/ijms222413384

**Published:** 2021-12-13

**Authors:** Paweł Kowalczyk, Dorota Sulejczak, Patrycja Kleczkowska, Iwona Bukowska-Ośko, Marzena Kucia, Marta Popiel, Ewa Wietrak, Karol Kramkowski, Karol Wrzosek, Katarzyna Kaczyńska

**Affiliations:** 1Department of Animal Nutrition, The Kielanowski Institute of Animal Physiology and Nutrition, Polish Academy of Sciences, 05-110 Jabłonna, Poland; p.kowalczyk@ifzz.pl (P.K.); marta.popiel@nutropharma.pl (M.P.); 2Department of Experimental Pharmacology, Mossakowski Medical Research Institute, Polish Academy of Sciences, 02-106 Warsaw, Poland; 3Analytical Group, Department of Analytical Chemistry and Biomaterials, Faculty of Pharmacy with Laboratory Medicine Division, Medical University of Warsaw, 02-097 Warsaw, Poland; patrycja.kleczkowska@wum.edu.pl; 4Military Institute of Hygiene and Epidemiology, 4 Kozielska St., 01-163 Warsaw, Poland; 5Department of Immunopathology of Infectious and Parasitic Diseases, Medical University of Warsaw, 02-091 Warsaw, Poland; ibukowska@wum.edu.pl; 6R&D Department Nutropharma LTD, Jedności 10A, 05-506 Lesznowola, Poland; marzena.kucia@nutropharma.pl (M.K.); ewietrak@nutropharma.pl (E.W.); 7Department of Physical Chemistry, Medical University of Bialystok, Kilińskiego 1 Str., 15-089 Białystok, Poland; kkramk@wp.pl; 8Department of Heart Diseases, The Medical Center of Postgraduate Education, Marymoncka 99/103, 01-813 Warsaw, Poland; wrzosekk@poczta.onet.pl; 9Department of Respiration Physiology, Mossakowski Medical Research Institute, Polish Academy of Sciences, 02-106 Warsaw, Poland

**Keywords:** mitochondria, mitochondrial diseases, oxidative stress, antioxidant therapy

## Abstract

The excessive formation of reactive oxygen species (ROS) and impairment of defensive antioxidant systems leads to a condition known as oxidative stress. The main source of free radicals responsible for oxidative stress is mitochondrial respiration. The deleterious effects of ROS on cellular biomolecules, including DNA, is a well-known phenomenon that can disrupt mitochondrial function and contribute to cellular damage and death, and the subsequent development of various disease processes. In this review, we summarize the most important findings that implicated mitochondrial oxidative stress in a wide variety of pathologies from Alzheimer disease (AD) to autoimmune type 1 diabetes. This review also discusses attempts to affect oxidative stress as a therapeutic avenue.

## 1. Introduction

The condition of the body associated with the excessive formation of reactive oxygen species (ROS) is known as oxidative stress. It indicates an imbalance between the production and accumulation of ROS, and the defence antioxidant systems [1]. ROS, kept at a low level, play a physiological role in intracellular signaling pathways [2]; however, when produced in an excessive amount, they become one of the main causes of cell and tissue damage. The latter results from direct harmful action on biological structures such as proteins, lipids, and nucleic acids [1,3]. Endogenous ROS are generated as by-products of oxygen metabolism, while exogenous oxidative stress can be evoked by environmental stressors such as ionizing or X-ray radiation, UV, pollutants, cigarette smoke, heavy metals, and certain drugs [4]. In the organism, chronic inflammatory processes aggravate oxidative stress and enhance the generation of ROS [5].

The most reactive free radicals are superoxide anion (O_2_^•−^) and the hydroxyl radical (OH•), a highly unstable species with unpaired electrons capable of initiating oxidation and generation further ROS, e.g., hydrogen peroxide (H_2_O_2_), peroxynitrite (ONOO−), and hypochlorous acid (HOCl) [4].

The main source of free radicals in the cell are the mitochondria. In addition to their primary role of ATP synthesis, mitochondria contribute to the biosynthesis of amino acids, nucleic acids, lipids, hemes, purines, and steroidogenesis [6,7]. They also control intracellular Ca^2+^ homeostasis and regulate thermogenesis, cell division, and programmed cell death [6,8]. During the intense oxidative metabolism, mitochondria generate and sequester reactive oxygen species, and approximately 1–2% of the molecular oxygen taken up by cells during physiological respiration is converted to ROS. In fact, the majority of free radicals and, mainly, superoxide anion are products of mitochondrial respiration, generated during electron flow in the mitochondrial electron transfer chain complexes I, II, and III [9,10]. The concentration of O_2_^•−^ in the mitochondrial matrix was estimated to be 5- to 10-fold higher than in the cytosol or nucleus [11]. Stimuli that induce oxidative stress in mitochondria are hypoxia, nutrient availability, cytokines, or changes in mitochondrial membrane potential [6,12]. In the case of an excessive formation of ROS or impaired mitochondrial antioxidant protection, ROS-induced damage to biomolecules (DNA, proteins and lipids) may compromise mitochondrial functioning and, together with the release of pro-apoptotic proteins from the mitochondrial intermembrane space, launch the activation of cell death. Thus, mitochondrial oxidative stress has been implicated in a wide variety of pathologies that will be discussed in this review.

### 1.1. Mitochondrial Oxidative Phosphorylation (OXPHOS)—A Source of Free Radical Formation

Mitochondria are cytoplasmic organelles that produce energy (adenosine triphosphate (ATP)) in the processes of oxidative phosphorylation. They are composed of a double membrane: an outer membrane separating the mitochondrion from the cytosol, and an inner membrane forming the mitochondrial cristae. The latter delineates the mitochondrial matrix that contains mitochondrial DNA (mtDNA) [13].

Oxidative phosphorylation takes place within the mitochondrial inner membrane where four large multi-subunit enzyme complexes are located; NADH: ubiquinone oxidoreductase (complex I), succinate dehydrogenase (complex II), coenzyme Q: cytochrome c reductase (complex III), cytochrome C oxidase (complex IV) [14].

These complexes compose the electron transport chain, where electrons are passed from one member of the transport chain to another in a series of redox reactions, from the reduced nucleotides, NADH and FADH2, to the electron acceptor oxygen, resulting in H_2_O generation. Energy released in these reactions is captured as a proton gradient, which is then used by ATP synthase (complex V) for the synthesis of ATP by the phosphorylation of ADP [14,15].

The main source of ROS formation in the mitochondrion, demonstrated in vitro, is electron leakage from respiratory complexes I, II, and III of the respiratory chain, mediating the one-electron reduction of oxygen to superoxide (O_2_^•−^) [14,16,17]. These data are confirmed by the work performed in vivo, where the main role in the production of ROS was demonstrated in complex I, where they are generated in the mitochondrial matrix, and complex III, which can produce ROS in the matrix or intermembrane space [14,18,19,20].

An additional process that generates a significant amount of superoxides is mitochondrial reverse electron transport (RET), which appears during electron leak when electrons are transferred back from complex II via ubiquinone to complex I, reducing NAD+ to NADH. ROS are produced by complex I when electrons circulate in the forward or reverse direction. During forward transport, electrons mainly leak to produce superoxide from complex I during the oxidation of NADH to NAD+. RET is favored by a high membrane potential that drives electron transport against the redox potential gradient of the electron transport chain [ETC]. This is conditioned by a highly reduced pool of coenzyme Q, evoked by electrons from respiratory complex II [17,18,19].

### 1.2. mtDNA and nDNA—Uniqueness of mtDNA

The mitochondrial genome exists in the form of the double-stranded, closed-circular DNA molecule, which contains of 37 genes. Twenty-four of the genes encode for two ribosomal RNAs, and 22 transfer RNAs that create the mitochondrial translation machinery. Thirteen of the genes encode proteins that instruct cells to produce subunits of enzyme complexes of the oxidative phosphorylation system [13,21,22]. 

Certain phosphorylation complex proteins are encoded within the nuclear genome, synthesized in the cytosol, and subsequently imported into the mitochondria. Thus, control of the mitochondrial energy production system occurs through two physically distinct genomes encoded by mtDNA and nuclear DNA (nDNA). The nuclear genome encodes most of the components or assembly factors of the OXPHOS system, and is also responsible for the maintenance and replication of mtDNA [23].

The uniqueness of mtDNA over nDNA that can lead to many diseases lies in its instability and susceptibility to attack by ROS due to the close proximity of mtDNA nucleoids to the inner side of the mitochondrial membrane and the electron transport chain, a major site of free radical production. The mitochondrial process of intracellular oxygen respiration generates reactive oxygen species, the major source of mutation in eukaryotes [9]. mtDNA is characterized by increased vulnerability to oxidative damage and mutations, which is 10 times greater than even nDNA. The causes of this are the lack of shielding histones, the limited number of mtDNA repair enzyme systems, and increased mtDNA supercoiling that changes mtDNA topology, which can cause genome instability [24,25]. Another reason for increased vulnerability of mtDNA to mutations can be its compactness: the lack of gene introns and the fact that some of the protein genes are overlapping [13]. Therefore, due to the high gene density, structural rearrangements and deletions in mtDNA are not well tolerated [13,26]. As a consequence, accumulated deletions or point mutations in important bioenergetic genes of mtDNA may disrupt the oxidative phosphorylation cycle, resulting in bioenergy disturbances that further affect cell and tissue survival.

## 2. Mutations in mtDNA and nDNA and Mitochondrial Diseases

Oxidative stress and ROS-induced DNA damage are potential causes of mutagenesis [23,27]. Although mtDNA has a greater tendency to accumulate mutations, those occurring in both nuclear and mitochondrial genes may lead to mitochondrial disorders that can be passed on to offspring through Mendel’s law inheritance for nDNA mutations and maternal inheritance associated with mtDNA mutations. Maternal inheritance means that mitochondria, along with the mtDNA, are inherited almost exclusively in the maternal-female line [28]. They originate mostly from unfertilized oocytes, because paternal mtDNA is selectively removed after fertilization in early zygote development [29]. The mitochondrial genome exhibits a high mutational burden, influenced by the haploidy of the mtDNA. The haploid uniparental inheritance usually associated with mitochondrial genetic bottlenecking makes recombination, to the extent that it occurs between identical genomes, irrelevant to the mutation removal process [29,30]. 

Another fundamental difference between mtDNA and nDNA is that the former occurs in thousands of copies within a cell. In most cases, the sequences of mtDNA molecules present in cells are identical, which is known as homoplasmy; however, frequent mtDNA mutations may result in heteroplasmia, that is, the co-occurrence of a mixture of molecules with different sequences. In addition, during mitotic and meiotic cell division, heteroplasmic alleles can change their proportion, which can vary between cells and tissues. Another important aspect is the fact that heteroplasmic mtDNA mutations can accumulate in body cells over time, the sum of which deteriorates the cell’s energy capacity [30,31]. Clinical symptoms of a mitochondrial disease depend on the percentage of heteroplasmia, the severity of the mutation, and tissue sensitivity to energy deficiency. Therefore, tissues with high metabolic requirements, such as the brain, heart, or muscles, are primarily affected and mitochondrial disorders typically include myopathies, encephalopathies, and neuropathies [23,31]. The example of the heteroplasmic mitochondrial disease caused by a point mutation in the gene encoding lysine tRNA in mtDNA is myoclonic epilepsy and ragged red fiber disease (MERRF), where the severity of the clinical phenotype is correlated with the proportion of mutant heteroplasmy corrected for age [31]. Another mitochondrial myopathy such as mitochondrial encephalomyopathy, lactic acidosis, and stroke-like episodes (MELAS) can be fatal when the heteroplasmy of a mutated gene is very high; whereas, at 10–30% heteroplasmy, the same mutation causes only maternally inherited diabetes and deafness [31]. Also, mutations in nDNA that codes mitochondrial OXPHOS subunits can result in primary mitochondrial diseases, which are a heterogenous multisystem group with a frequent lack of genotype–phenotype correlations and a complicated diagnosis [32,33]. They can be associated with defects in nDNA genes encoding structural subunits, assembly and translational factors impairing energy metabolism by inactivating an OXPHOS polypeptide, disrupting antioxidant defenses, mtDNA replication and repair, or mitochondrial quality control [14]. For example, nuclear mutations involving subunits of oxidative phosphorylation complex I, the largest enzyme complex of the oxidative phosphorylation chain, can lead to Leigh syndrome, lactic acidosis with coexisting cardiomyopathy or leukodystrophy [34].

The exact classification and clinical manifestations of mitochondrial diseases associated with mutations in mt- and nDNA are described and summarized in detail elsewhere, according to [23,32,34,35].

## 3. Diseases Associated with Impaired ROS Generation in Mitochondria

### 3.1. Neurological Diseases

Neurodegenerative diseases such as Alzheimer’s disease (AD), Parkinson’s disease (PD), and amyotrophic lateral sclerosis (ALS) are age-related conditions characterized by significant changes in mitochondrial structure and function associated with free radicals generation [36].

Increased levels of free radicals and higher oxidation of macromolecules including mtDNA have been observed in Alzheimer’s disease (AD) human brains and in various animal models [37]. What is more, free radicals have been shown to increase the activity of β- and γ-secretases, enzymes responsible for amyloid β generation from amyloid precursor protein [38]. Further, Nonomura et al. [39] demonstrated that oxidative damage is quantitatively greatest early in the AD and decreases with dementia progression and amyloid β plaque deposition. It has been proposed that mitochondrial oxidative stress damages mtDNA encoding electron transfer chain subunits, which negatively affects ATP production and calcium homeostasis, and exacerbates oxidative stress. The latter, in turn, increases amyloid β deposition and leads to further consequences of neuronal dysfunction, neurodegeneration, and cognitive impairment in AD [36,40,41,42].

It is still not clear whether mitochondrial dysfunction plays a direct role in the initiation of AD according to the “mitochondrial cascade hypothesis” or is, rather, a consequence of amyloid β accumulation. Indeed, Reddy [43] suggested that progressive mitochondrial damage leading to disease progression is caused by β-amyloid entry into mitochondria, triggering the production of free radicals. Oxidative damage and neuroinflammation have been shown to correlate with Alzheimer’s disease progression [41,44]. A synergistic role of both pathways is also possible [41,45]. It is certain, however, that many of the therapies targeting mitochondrial dysfunction in neurodegeneration and cognitive dysfunction in AD rely on the application of antioxidants and a reduction in free radical levels [42].

Mitochondrial damage closely related to oxidative stress seems to play an important role in the pathogenesis of Parkinson’s disease (PD) [36,46,47]. At the cellular level, PD is caused by both the overproduction of reactive oxygen species and changes in dopamine metabolism, as well as alteration in the mitochondrial electron transporter chain function in the neurons of substantia nigra [48]. The involvement of oxidative stress in dopaminergic cell degeneration was indicated further by the increased oxidative damage to mtDNA noted in PD neurons of substantia nigra [49,50,51]. Even mutations in genes coding proteins linked to PD such as DJ-1, parkin, PINK1, alpha-synuclein, and LRRK2 affect mitochondrial function and integrity, causing enhanced ROS generation and vulnerability to oxidative stress [48,52]. Currently, the role of antioxidant neurotrophic strategies in PD treatment is emphasized. One of them is the proposal to combine antioxidant therapy with stem cell therapy to reduce damage and induce repair of dopaminergic neurons for the treatment of Parkinson’s disease [48,53]. 

Oxidative stress exacerbating damage to mitochondria has been also identified as one of the factors involved in demyelination, axonal and neuronal death in multiply sclerosis (MS), and motoneuron death in amyotrophic lateral sclerosis (ALS) [54,55,56]. Undoubtedly, an inflammatory process engaged in oligodendrocyte pathology that activates and recruits lymphocytes, macrophages, and microglia is able to generate vast quantities of oxidizing radicals contributing to MS tissue injury [57]. In the case of ALS pathology, the involvement of ROS is supported by the elevated free radical levels in the cerebrospinal fluid, serum, and urine of patients with sporadic and familial forms of ALS [55,58,59]. In addition, in familial ALS, altered reactivity of superoxide dismutase, responsible for the clearance of reactive oxygen species, is reported [60]. As shown by Petrozziello et al. [61], oxidative stress in ALS causes mitochondrial fragmentation and dysfunction. Unfortunately, clinical trials of antioxidant therapy appear to be unsuccessful despite beneficial effects in animal models [62]. Recently, the reduction of oxidative stress damage has been shown to effectively prolong animal survival time and reduce brain pathological symptoms in a mouse model of ALS [62,63]. 

The causes of schizophrenia are as yet undetermined. One hypothesis points to oxidative stress as the contributing factor to the pathophysiology of the disease [1,64]. This is supported by decreased levels of antioxidants and augmented oxidative stress markers in schizophrenic patients [65,66,67,68]. Significantly reduced glutathione (antioxidant) levels have been reported in magnetic resonance spectroscopy in the cerebral cortexes of living patients [69], but also in post-mortem examination [70]. Computer tomography scans showing brain atrophy in chronic schizophrenic patients revealed strong correlation between brain pathology and low glutathione peroxidase activity in platelets [1,71]. In addition, the oxidative imbalance in schizophrenia was paralleled by increased severity of negative symptoms of the disease [66]. Cuenod and colleagues [72] emphasize the role of complex mechanisms of oxidative stress and its modulation in the pathophysiology of schizophrenia, and attribute a major role to dysregulation of redox mechanisms, disruption of mitochondrial bioenergetics, and neuroinflammation in the development of oxidative stress during neurodevelopment. The role of one of the forms of oxidative stress, the so-called carbonyl stress, is currently being studied in the pathophysiology of schizophrenia. Hara et al. [73] indicate that this stress causes mitochondrial damage, lowers mitochondrial membrane potential, and hinders aerobic respiration processes. Even genetic predisposition linked to mitochondrial function and subsequent oxidative stress has been found; gene cacna1c is considered as a strong genetic risk factor for the development of affective disorders [74]. Although the evidence is inconsistent, there are studies demonstrating the efficacy of antioxidant therapies in the treatment of schizophrenia that support the hypothesis that oxidative stress plays an important role in its development [64].

### 3.2. Neurodevelopmental Disorders

Oxidative stress induced by prenatal exposure to toxic chemicals is regarded as a key factor in the occurrence of neurodevelopmental disorders [75]. In the case of autism mitochondrial abnormality, augmented oxidative stress and decreased antioxidant capacity have been reported in autistic persons, all of which may be responsible for neuroinflammation and autism pathology [76,77]. Recent analysis of blood samples from children with autism spectrum disorders revealed reduced total plasma peroxidase and total antioxidant capacity, resulting in an imbalance in the oxidant/antioxidant ratio and abnormalities in neuronal transduction [78]. Zawadzka et al. [79] showed that impaired brain development is a consequence of inflammatory processes inducing oxidative stress and mitochondrial damage, which in turn exacerbate oxidative stress, triggering further cellular damage. In support of the role of oxidative stress in autism pathology, studies using n-acetylcysteine or other antioxidants have reported a reduction in some autistic behaviors in children, such as irritability and hyperactivity [76,80,81]. 

### 3.3. Autoimmune Diseases

Another group of diseases whose pathomechanism may involve mitochondrial dysfunction causing oxidative stress are T cell-mediated autoimmune diseases such as type 1 diabetes (T1D), multiple sclerosis (MS), rheumatoid arthritis (RA), and systemic lupus erythematosus (SLE) [82,83]. The autoreactive T cells that recognize systemic or organ-specific self-antigens, responsible for autoimmunity, are susceptible to ROS that are engaged in their differentiation, effector responses, and inducing proinflammatory cytokine release [82,84]. The latter triggers inflammation involved in the pathomechanism of autoimmune disorders, resulting in oxidative stress and damage to cellular macromolecules. Oxidative stress and inflammation are closely related. Mitochondrial-derived ROS via the oxidation of biomolecules or structural modification of proteins and genes may start signaling cascades, leading to inflammatory processes. ROS-activated transcription factors and pro-inflammatory genes induce inflammation and recruitment of immune and inflammatory cells to the site of oxidative stress. Activated immune cells generate ROS at the site of inflammation, amplifying oxidative stress and tissue injury [5,85,86,87]. 

In SLE, patients show increased ROS in T cells as well as more oxidized lipoproteins, which can lead to vascular inflammation and atherosclerosis [88]. Another pathway of action of ROS on the development of an autoimmune SLA is the damage of DNA, which becomes a major antigenic target for autoantibodies [89].

In T1D, profound metabolic changes occur during insulin deprivation including an increase in basal energy expenditure and reduced mitochondrial function [89,90]. Sustained hyperglycemia induces increased ROS production, and systemic oxidative stress has been confirmed at early onset of T1D, as well as its increase in early adulthood [89,91]. Indeed, mitochondria-derived free radicals has been demonstrated to contribute to the process of immune-mediated beta-cell destruction via the induction of cytokine toxicity in T1D [89,92]. Another reason is that beta-cells exhibit insufficient antioxidant defense, which is associated with low expression of antioxidant enzymes in islets [84].

The chronic oxidative stress in the RA is characterized by a significant increase in mitochondrial ROS production [93]. It contributes to joint damage, playing the role of messenger in inflammatory and immunological cellular response including activation of the NLRP3 inflammasome, which produces cytokines linked to RA symptoms [83].

### 3.4. Kidney and Lung Diseases

Other diseases associated with mitochondrial oxidative stress and inflammation are chronic kidney disease (CKD) and chronic obstructive pulmonary disease (COPD). Mitochondrial dysfunction, such as decreased mtDNA, and ATP production, as well as the loss of mitochondrial membrane potential, related to increased mitochondrial ROS, has been shown to precede kidney injury and further contribute to the development and progression of CKD, characterized by a decrease in the number of active nephrons [94]. Excess ROS present early during CKD progression and contribute to inflammatory process in the renal parenchyma via inflammatory cell recruitment and proinflammatory cytokine production, leading to endothelial impairment and atherosclerosis [95]. Interestingly, the mechanism of nephrotoxicity of some drugs (cyclosporine, gentamycin) has been demonstrated to involve oxidative stress induction and lipid peroxidation [96].

A leading cause of COPD is cigarette smoking. Cigarette smoke, particulate matter, and noxious gases including ozone are major exogenous sources of ROS that challenge respiratory epithelial cells and injure small airways and lung parenchyma directly or indirectly by increasing inflammation [97,98,99]. Nevertheless, inflammation and oxidative stress are inextricably linked. Indeed, oxidative stress-induced tissue damage can trigger inflammation and immune responses, which in turn can enhance ROS production [5,100].

Airway smooth muscle and bronchial biopsies from COPD patients showed increased mtROS production and decreased antioxidant enzymes compared to healthy control subjects [101,102]. Further, impaired redox regulation associated with cellular ageing has been described to contribute to the development and acceleration of COPD pathogenesis via enhanced inflammation, protease–anti-protease imbalance, and cellular apoptosis [103].

### 3.5. Cardiovascular Diseases (CVDs)

ROS are considered as one of the major causative factors leading to atherosclerosis development. Oxidative stress contributes to atherosclerotic plaque formation via induction of endothelial dysfunction, vascular inflammation, and accumulation of oxidized low-density lipoprotein [104]. All these lead to lesion formation and accumulation of macrophages, which, apart from producing ROS, phagocytize oxidized lipoproteins and transform into foam cells, components of atherosclerotic plaque [105,106]. Oxidative stress markers have been shown to be elevated in patients suffering from cardiovascular diseases such as hypertension [107,108] and heart failure, whereas its increase in cardiomyocytes is correlated with the development and the progression of maladaptive myocardial remodeling [109,110,111]. Cardiac dysfunction associated with metabolic syndrome comprising of diabetes, high blood pressure, and obesity is actually due to enhanced oxidative stress causing damage of mitochondria, the activation of mitochondria apoptotic signaling pathways, and cardiomyocyte contractile dysfunction [112].

Interestingly, numerous studies indicate that the protective nature of estrogen against cardiovascular disease risk in premenopausal women is due to its oxidative stress-inhibitory properties [113].

### 3.6. Cancer

Elevated ROS mutagenicity results from the induction of genetic instability evoked via increasing receptor and oncogene activity, stimulation of oxidative enzymes or growth factor-signaling pathways involved in regulation of DNA repair, cell proliferation, apoptosis, and tumorigenesis [114,115,116].

As mentioned earlier, excess ROS can also directly damage DNA by causing single- and double-strand nucleic acid breaks and by forming an oxidized derivative of deoxyguanosine, 8-Oxo-2′-deoxyguanosine, which contribute to carcinogenesis through promoting mutagenesis [116]. Consequently, mutations in mtDNA, reduced mtDNA content, and mutations in nuclear genes can irreversibly damage mitochondrial oxidative phosphorylation. The latter leads to mitochondrial dysfunction and further genetic instability in the nuclear genome, and is one of the proposed causes of cancer [116,117].

Not surprisingly, oxidative stress may be responsible for the onset and development of various types of cancer from hepatocellular carcinoma, breast cancer, and lung cancer to brain tumors [118,119,120]. ROS have been also shown to induce DNA hypermethylation, which can affect the tumor phenotype [114,120].

Oxidative stress can act on cancer cells in two ways, which should be taken into account in the design of anti-cancer drugs targeting ROS. In physiological amounts, ROS contribute to further cancer growth by transducing signals for cell proliferation, migration, and angiogenesis, whereas severe oxidative stress may produce a deleterious effect through the induction of cell-cycle arrest and apoptosis [116]. However, cancer cells are able to resist excessive intracellular ROS by activating the transcription factor and nuclear erythroid 2-related factor (NRF2) responsible for antioxidant enzymes transcription, promoting cancer cell survival [116,120].

All disease entities induced by mitochondrial damage are presented in Figure 1.

## 4. Oxidative Stress as a Therapeutic Target

Since oxidative stress may contribute to, or is the major cause of, the development and the progression of a great variety of diseases and disorders, it has garnered interest as a potent therapeutic target. Obviously, the organism has a self-defense mechanism that, in the case of oxidation, involves antioxidants; either detoxifying oxidants or preventing oxidative damage through the direct/indirect prevention of •OH formation. One of the examples is superoxide dismutase (SODs) enzymes, characterized by the ability to remove H_2_O_2_ and lipid hydroperoxidases, though these properties are different for extracellular SOD (SOD3) [121].

SODs are known to be the most important endogenous antioxidants, which encompass three types of enzymes, i.e., SOD1, or CuZn-SOD, found almost exclusively in intracellular cytoplasmic spaces; SOD2, or manganese Mn-SOD, existing in mitochondrial spaces; finally, extracellular SOD3, or EC-SOD [122]. Considering that they catalyze the dismutation (conversion) of O_2_^−^ to H_2_O_2_ and O_2_, in order to fight against oxidative stress-induced pathology, an increased level of SODs is required. Indeed, an increase in vascular permeability or reperfusion injury after ischemia is related to SOD deficiency [123]. Also, in a paper by Isogawa et al. [124], a reduced SOD activity was found in patients with cardiovascular disease (CVD). Similar observations were performed by Soto et al. [125] in hypertensive patients with aortic dilatation, as well as in the paper by Gupta et al. [126], describing a decrease in SOD activities in serum samples of breast cancer patients in comparison to healthy controls. Nonetheless, there are also several papers indicating opposite information. In fact, it has been revealed that SOD1 expression was increased within the neuropathologic lesions in brains of AD patients [127] as well as in AD-transgenic mice [128]. Intriguingly, similar increases were shown in patients with amyotrophic lateral sclerosis [129] and Down syndrome [130] who carry an extra copy of the SOD1 gene. Of interest here is the fact that, in the case of AD, the expression of SOD1 was found to be increased while its activity was reduced. The explanation to this was that, while affected neurons and reactive glia do respond to oxidative stress by increasing the expression of the enzymes, the majority of the newly synthesized enzymes may be rendered inactive by oxidation or by other biochemical alterations intrinsic to the aging or neuropathologic process [127].

Although the importance of SOD in oxidative stress is undeniable, unfortunately, the clinical application of SOD as a therapeutic agent has been limited due to its low half-life and, thus, its extremely rapid plasma clearance time, as well as its instability [131].

Therefore, based on the knowledge of the potency of naturally existing compounds, several synthetic analogs were developed in order to obtain more improved properties (e.g., half-life, smaller size) and, thus, to provide more effective therapies. With reference to SOD, small molecule catalytic antioxidants, known as SOD mimetics, that possess similarity in function to the SOD native enzyme, were designed and developed. For instance, the use of MN porphyrin—MnTnBuOE-2-PyP(5+)—a potent SOD mimetic, resulted in the enhancement of carbenoxolone-mediated TRAIL-induced apoptosis in even the most malignant tumor of the brain [132]. Also, GC4419, a Mn(II)-containing pentaazamacrocyclic SOD mimetic, was shown to remove superoxide anions in a selective manner, with no interaction towards other oxidants [133]. Intriguingly, this compound revealed its therapeutic effect either in cancer [134] or inflammation, but also in joint disease [135].

Glutathione peroxidases (GPx) are the second-most effective first line antioxidant enzymes that were widely investigated for the treatment of conditions resulting from oxidative damage. These compounds (GPx1–GPx8) are known for their capability to break down hydrogen peroxide into water with the use of reduced glutathione [136]. However, it should be noted that GPx activity is highly dependent from the existence of selenium, thus GPx is mostly referred to as a selenocysteine peroxidase [137]. GPx (GPx-3) is the most important selenoenzyme in the detoxification of ROS in human plasma. Importantly, in the early stage of colorectal cancer, both GPx-1 and GPx-3 expressions were found to be decreased, whereas GPx-2 expression increased [138]. Additionally, for instance, GPx-1 overexpression combined with selenium supplementation protected mammalian cells against ultraviolet (UV)-induced DNA damage [139].

Unfortunately, the effectiveness of enzymatic antioxidants has recently been questioned, as some of the papers deliver different results. In fact, for instance, SOD enzymes may also be responsible for the occurrence of some diseases, as approximately 20% of the familial cases of Amyotrophic Lateral Sclerosis appear to arise as a consequence of mutations in the copper/zinc superoxide dismutase gene SOD1 [140,141]. Similar unexpected findings were found for SOD2 (manganese mitochondrial SOD, MnSOD), for which high levels were found in tumor tissues, particularly in stages II and III of malignancy versus stage I or precancerous stage [142]. Of note, these changes were reported to be strictly related with the aerobic glycolysis in tumor cells as a consequence of SOD2-induced activation of AMPK, which further led glucose metabolism via glycolysis [143]. Nonetheless, it has been suggested that SOD2 expression in human cancer might be stage- and/or tumor-type-dependent [144]. Similar observations were found in the case of GPx. Indeed, Gouaze et al. [145] found that the overexpression of GPx-1 resulted in significant resistance of breast cancer T47D cells to doxorubicin. However, it is obvious that many cancer cells upregulate antioxidant enzymes to make sure ROS levels do not become high enough to cause cell death [145,146].

Apart from the antioxidant enzyme system, the mammalian organism is also equipped with so-called non-enzymatic antioxidants which include uric acid (UA), glutathione, bilirubin, melatonin, etc. Additionally, exogenous non-enzymatic antioxidants can be distinguished, i.e., vitamin E, vitamin C, and vitamin A (these may be provided with a specific diet). Glutathione (GSH), a thiol-containing peptide that exists intracellularly in either an oxidized (glutathione disulfide, GSSG) or the thiol-reduced (GSH) form, is one of the most studied and most important antioxidants. Indeed, apart from its impact on immune system function and inflammation, its reduced levels contribute to the onset and progression of many diseases [147] for which oxidative stress has been proposed [148] to be one of the pathogenic mechanisms [149]. Such an antioxidant activity is seen in both in vitro and in vivo studies. A great example is a recent paper by Kwon et al. [150], who demonstrated GSH to inhibit a H_2_O_2_-induced cytotoxic effect in the monocyte/macrophage RAW 264.7 cell line, though there are other papers previously published on the topic [151,152,153,154]. Moreover, GSH genes, especially GSTM2 [154], were proposed to be involved in the protection against neurodegenerative diseases. In addition, the manipulation of endogenous GSH concentrations can alter cellular responses to oxidant injury. For instance, the first who revealed the relationship between GSH depletion and increased susceptibility to oxidant injury were Meredith and Reed, who have presented that the onset of ethacrynic acid-mediated cellular injury in hepatocytes in vitro was correlated with the depletion of mitochondrial GSH [155].

Although the role of uric acid in oxidative stress is still ambiguous, it can be shown that its antioxidant properties were revealed in conditions affecting the central nervous system, i.e., Parkinson’s disease or multiple sclerosis [156]. Its beneficial effect was also observed in mice induced with allergic encephalomyelitis (EAE) [156]. Intriguingly, similar action was also presented in acute stroke; however, not in chronic ones [157], as it was found that chronic increase in uric acid level may be associated with the risk of stroke. Such unexpected behavior was further confirmed by the fact that uric acid was found to form a great variety of free radicals via reactions with different oxidants [158,159].

Bilirubin, a byproduct of hemoglobin breakdown, is characterized by its antioxidant potential against peroxyl radicals (ROO^·^) and hydrogen peroxide [160]. It was recently presented as a dose-dependently-acting compound that exerts its anticancer activity against colon cancer (HRT-18) and hepatocellular carcinoma HepG2 and NIH/3T3 in vivo [161].

Melatonin, *N*-acetyl-5-methoxytryptamine, is well known for its mood regulatory and sleep–wake cycle regulatory properties [162]. However, it is also responsible for maintaining the oxidant/antioxidant balance [163]. Interestingly, the mechanism by which melatonin induces its antioxidative effect is complex. Indeed, it was found to increase both the expression as well as the activities of several antioxidant enzymes (i.e., SOD, GPX), together with non-enzymatic glutathione. It may also interact in a synergistic manner with antioxidants or, finally, enhance the efficiency of the mitochondrial electron transport chain [164,165,166,167]. Melatonin’s undeniable role as an antioxidant was also observed in animals and human patients suffering from neurodegenerative diseases. In fact, recently, a paper by Muhammad et al. [168] presented melatonin to attenuate scopolamine-induced synaptic dysfunction and memory impairments. These actions were exerted by ameliorating oxidative brain damage, stress kinase expression, neuroinflammation, and neurodegeneration. Moreover, an application of melatonin prevented the death of neuroblastoma cells exposed to Aβ peptide, as well as inhibited the formation of amyloid fibrils [169,170].

Although melatonin seems to be a promising drug, effective in the treatment of damages induced by oxidative stress or even in the prevention of those damages, we do not know how this hormone acts in high doses or during long-term treatment. Reported short-term side effects including dizziness, headache, nausea, and sleepiness, although they seem to be mild [169,170,171]. 

Research from recent years has shown that some vitamins, e.g., vitamin D, vitamin C, minerals such as selenium and zinc, and proteins such as lactoferrin [172,173,174], have been proven to act positively on immunity and reduce the harmful effects of oxidative stress. They can affect the functioning and effectiveness of the immune system in reducing and removing pathogenic viruses, including the new, dangerous SARS-CoV-2 virus responsible for COVID-19. Preventive measures based on regular consumption of dietary supplements composed of antioxidants or elements with antioxidant properties such as vitamin D, vitamin C, selenium, zinc, and lactoferrin may be an important method to enable public health control in breaking transmission chains [174,175,176,177,178,179]. All antioxidants that protect from mitochondrial damage are presented in Figure 2.

## 5. Conclusions

Given the presence of enhanced mitochondrial oxidative stress in many pathologies, it may be a causative factor in the development of a wide variety of conditions such as neurodegenerative and autoimmune diseases, and diseases of the lung, heart, kidney, or cancer. Recent reports have also indicated a potential role of oxidative stress in the pathology of schizophrenia or autism. Therefore, a therapeutic pathway for their prevention or treatment of these may be the reduction of oxidative stress. Although the effectiveness of some oxidative stress relieving therapies has been confirmed, the data are often conflicting or inconclusive. Sometimes there is a lack of an effective way to prevent the generation of—or to reduce the actually existing—oxidative stress. It is important to mention that oxidative stress is a complex mechanism involving different pathways, highly specific to a particular cell type, as is the action of oxidants and antioxidants. For example, severe oxidative stress can be harmful to cancer cells, and, therefore, inhibiting it may be detrimental. Antioxidants are also known to prevent NF-κB activation; however, they also unexpectedly increase binding of the active form, leading to unknown effects that can sometimes be counterproductive. Therefore, antioxidant therapies should be designed with caution.

## Figures and Tables

**Figure 1 ijms-22-13384-f001:**
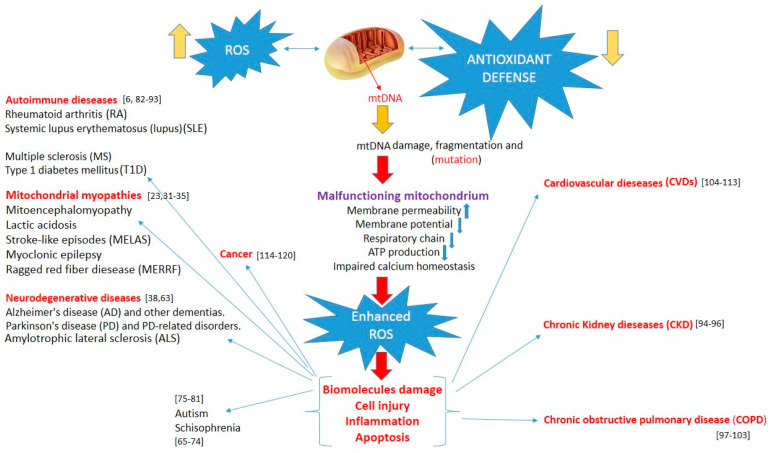
Increased reactive oxygen species, overwhelming antioxidant defenses, induce mtDNA damage, and mitochondrial dysfunction lead to enhanced oxidative stress. This, in turn, can induce biomolecule and cell damage, apoptosis, and inflammation, triggering various pathologies.

**Figure 2 ijms-22-13384-f002:**
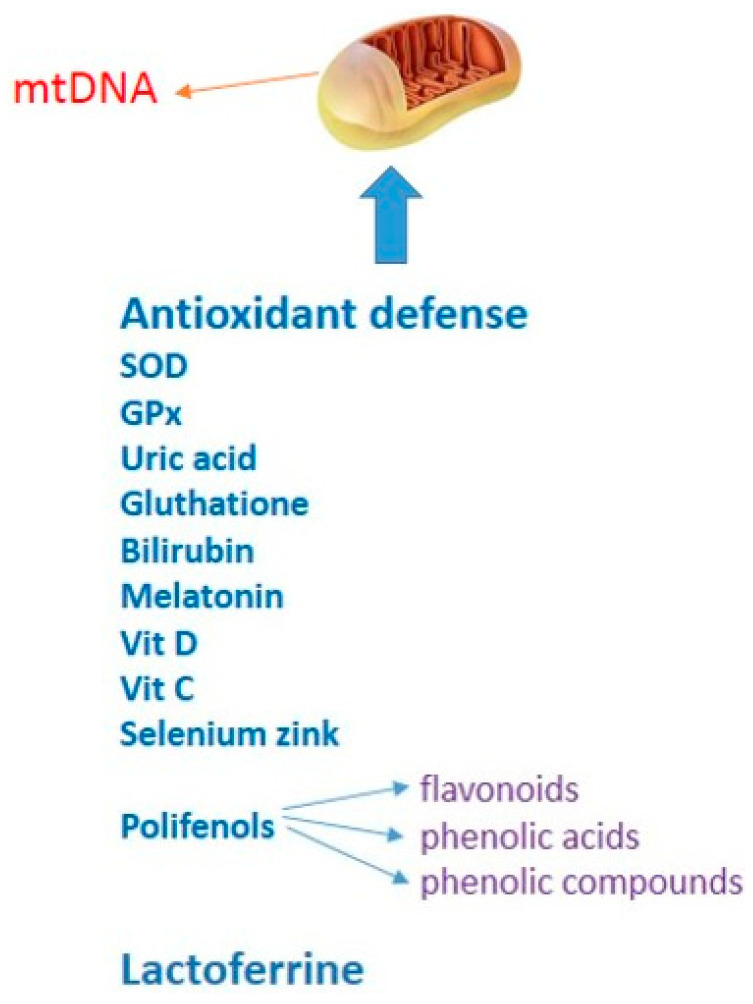
Antioxidant defense against mitochondrial damage.

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
