# Peer review of "Mitochondrial Oxidative Stress—A Causative Factor and Therapeutic Target in Many Diseases"

_ijms, 2021, doi:10.3390/ijms222413384_

Round 1

Reviewer 1 Report

1. The reviewing on the relationship between mitochondrial dysfunction and disease is weak, although the authors entitled the manuscript “Mitochondrial Oxidative Stress—A Causative Factor in Many Diseases”. The current title expresses only half of the content of the manuscript. The authors may select to change the title or to delete the part of therapeutic agents, which occupies the latter half of the manuscript, and strengthen the review on the relationship between mitochondrial dysfunction and disease.
2. Lane 72-87: The reviewer suggests to rewrite the Introduction part simple. Since it can be expected that the readers of this manuscript have a basic knowledge about mitochondria, the authors should not spend words on the unnecessary descriptions. The paragraph of “1.1 Mitochondria-Oxidative Phosphorylation” is not necessary and to be deleted. Instead, the authors can cite a few great reviews on mitochondria in the Introduction par.
3. Lane 88-102: The paragraph of “1.2 Mitochondria- mtDNA and nDNA” should be rewritten. This paragraph should present the readers the uniqueness of mtDNA and why those uniqueness can be a cause of the disease.
4. Lane 103-151: In the paragraph of “2 Mutation on mtDNA and Mitochondrial Diseases”, the authors might add the description and discussion of what kinds of disease “mitochondrial oxidative stress” onto mtDNA (and nDNA) can induce.
5. Lane 103-151: The authors use “mtDNA and nDNA”, “mt- and nDNA”, “mitochondrial DNA and nuclear DNA” and notation should be unified. 
6. Lane 103-151: The authors should point out the importance of the fact that nDNA is diploid (2n) and mtDNA is haploid (n), and it could be one of underlying causes of the disease by increasing the mutation rate of mtDNA. 
7. Lane 116: The sentence that “because sperm mtDNA is removed by ubiquitination in early zygote development” should be rewritten, since mtDNA can not be ubiquitinated. 
8. Lane 118-136: It would be better to introduce specific disease names first and discuss heteroplasmic mtDNA mutations as a mechanism, to help the readers understand the disease machinery.
9. Many unnecessary line breaks: Lane 144-145, Lane 264-265, Lane 346-347, Lane 359-360, Lane 374-375.
10. Lane 176: The format of citations is incorrect. It should be [40].
11. Lane 159-196: Descriptions on AD, PD, and ALS are poor. AD, PD, and ALS are the diseases that the readers of this manuscript are most likely interested in, and the poor of citations (8 for AD, 9 for PD, and 8 for ALS) weakens the value of this review manuscript. 
12. Lane 197-201: The relationship between Schizophrenia and mitochondrial oxidative stress should be of interest to many readers, and the review and discussion in this manuscript are too weak to meet their expectations.
13. Lane 202-207: The relationship between neurodevelopmental disorder and mitochondrial oxidative stress should be of interest to many readers, and the review and discussion in this manuscript are too weak to meet their expectations.
14. Lane 208-226: Many readers will be interested in the relationship between autoimmune disease and mitochondrial oxidative stress, and the review and discussion in this manuscript are too weak to meet their expectations.
15. Lane 208-226: In this section, t he authors may discuss the machinery of disease, emphasizing the significance of ROS in the regulation of immune responses. Indeed, oxidative stress-induced tissue damage can trigger inflammation and immune responses, which in turn can enhance ROS production.
16. Lane 208-226: The reviewer requests the authors to review related literatures and add a detailed description and discussion of how mitochondrial dysfunction leads to inflammation, since this is a hot area of basic medicine research and that many readers should be interested in.
17. Lane 209-215: It lacks a review and a discussion in terms of metabolic activity in diabetes.
18. Lane 279: Figure 1 should appear earlier.
19. Lane 282-392: If the authors will not change the title, this part (“4. Oxidative stress as a therapeutic target”) should be deleted and the review and the discussion of the relationship between mitochondrial dysfunction and disease should be strengthened.
If the authors will change the title, 
Lane 303-316 are unnecessary.
Lane 317-327 are unnecessary.
Lane 328-341 might be unnecessary.
20. Lane 379-392: The reviewer cannot understand the intent of the discussion and ask the authors to rewrite it. Citing the 27 references (123-140) at one time is highly unusual and no use for review.
21. Figure 2 is not necessary for the reader.
22. About “Zilavir”: If the authors have received funding or budget from the manufacturer of the drug or other related sources in the past, the authors should describe it in the Conflicts of Interest section. In the absence of such a past, the reviewer believes that specific product names should be avoided in the text of academic papers and the chemical names of the product should be described.
23. Lane 396-404: The reviewer believes that it is inappropriate to cite references in the Conclusion part. The reference 121 and 122 are not best ones and could be replace with more suitable ones.
24. In “Author Contribution”: The reviewer dose not understand why there are contributions in methodology, software validation, and resources in a review article. KK (Karol Kramkowski) and KW (Karol Wrzosek) have contributed only in Supervision. Joanna Mikoda (JM) has no contribution in this manuscript and can not be an author of this manuscript. The statement by the authors that all 10 (11-JM) of the authors were in charge of Supervision is unacceptable as a fact, and the reviewer have to distrust the statement by the authors.

Author Response

Reviewer 1

Thank you very much for the apt suggestions that contributed to the improvement of the quality of the manuscript. All comments and suggestions have been taken into account and marked in red or a block in green.

1. The reviewing on the relationship between mitochondrial dysfunction and disease is weak, although the authors entitled the manuscript “Mitochondrial Oxidative Stress—A Causative Factor in Many Diseases”. The current title expresses only half of the content of the manuscript. The authors may select to change the title or to delete the part of therapeutic agents, which occupies the latter half of the manuscript, and strengthen the review on the relationship between mitochondrial dysfunction and disease.

-According to Referee suggestion the title has been changed to: “Mitochondrial Oxidative Stress—A Causative Factor and Therapeutic Target in Many Diseases”.

2. Lane 72-87: The reviewer suggests to rewrite the Introduction part simple. Since it can be expected that the readers of this manuscript have a basic knowledge about mitochondria, the authors should not spend words on the unnecessary descriptions. The paragraph of “1.1 Mitochondria-Oxidative Phosphorylation” is not necessary and to be deleted. Instead, the authors can cite a few great reviews on mitochondria in the Introduction par.

-The paragraph is very short. Since the other reviewers think it is important and suggest to add some information, we have decided to leave it.

3. Lane 88-102: The paragraph of “1.2 Mitochondria- mtDNA and nDNA” should be rewritten. This paragraph should present the readers the uniqueness of mtDNA and why those uniqueness can be a cause of the disease.

- This has been done according Reviewer suggestion.

4. Lane 103-151: In the paragraph of “2 Mutation on mtDNA and Mitochondrial Diseases”, the authors might add the description and discussion of what kinds of disease “mitochondrial oxidative stress” onto mtDNA (and nDNA) can induce.

-As yet, it is not known which mutations in mtDNA and nDNA causing mitochondrial diseases are specifically evoked with “mitochondrial oxidative stress”. It is well  known that oxidative stress induces mutations that may be pathogenic therefore it is considered as a potential cause of many disorders, which is now stated in the text(…)

5. Lane 103-151: The authors use “mtDNA and nDNA”, “mt- and nDNA”, “mitochondrial DNA and nuclear DNA” and notation should be unified. 

-This has been unified to mtDNA and nDNA

6. Lane 103-151: The authors should point out the importance of the fact that nDNA is diploid (2n) and mtDNA is haploid (n), and it could be one of underlying causes of the disease by increasing the mutation rate of mtDNA. 

- Thank you for your suggestion, this has been added.

7. Lane 116: The sentence that “because sperm mtDNA is removed by ubiquitination in early zygote development” should be rewritten, since mtDNA can not be ubiquitinated. 

 - This has been rewritten

8. Lane 118-136: It would be better to introduce specific disease names first and discuss heteroplasmic mtDNA mutations as a mechanism, to help the readers understand the disease machinery.

- This is not a systematic review. The authors did not aim to describe in detail the diseases associated with heteroplasmic mtDNA, but only to give some examples. These diseases are described in detail in other papers referenced by us. However, the phenomenon of heteroplasmia has been explained by us.

9. Many unnecessary line breaks: Lane 144-145, Lane 264-265, Lane 346-347, Lane 359-360, Lane 374-375.

- They have been deleted.

10. Lane 176: The format of citations is incorrect. It should be [40].

-It has been corrected.

11. Lane 159-196: Descriptions on AD, PD, and ALS are poor. AD, PD, and ALS are the diseases that the readers of this manuscript are most likely interested in, and the poor of citations (8 for AD, 9 for PD, and 8 for ALS) weakens the value of this review manuscript. 
12. Lane 197-201: The relationship between Schizophrenia and mitochondrial oxidative stress should be of interest to many readers, and the review and discussion in this manuscript are too weak to meet their expectations.
13. Lane 202-207: The relationship between neurodevelopmental disorder and mitochondrial oxidative stress should be of interest to many readers, and the review and discussion in this manuscript are too weak to meet their expectations.
14. Lane 208-226: Many readers will be interested in the relationship between autoimmune disease and mitochondrial oxidative stress, and the review and discussion in this manuscript are too weak to meet their expectations.

-I wanted to emphasize that this is not a systematic review, rather a short review giving a quick overview of oxidative stress involvement in different pathologies. There are already overviews for several of these diseases  (AD, PD and ALS) and we refer the reader wishing to explore the topic in more depth to them. It was not our purpose to write the same thing that has already been written. As for some diseases, like schizophrenia or autism, there is not much information, but that is beyond our control. We just wanted to include the most important information pointing to relationship of described diseases to mitochondrial oxidative stress. However, some new information has been added to these disorders.

15. Lane 208-226: In this section, the authors may discuss the machinery of disease, emphasizing the significance of ROS in the regulation of immune responses. Indeed, oxidative stress-induced tissue damage can trigger inflammation and immune responses, which in turn can enhance ROS production.

-This has been added to the text.

16. Lane 208-226: The reviewer requests the authors to review related literatures and add a detailed description and discussion of how mitochondrial dysfunction leads to inflammation, since this is a hot area of basic medicine research and that many readers should be interested in.

-Oxidative stress as mitochondrial dysfunction leading to inflammation has been added to the text.

17. Lane 209-215: It lacks a review and a discussion in terms of metabolic activity in diabetes.

-This has been added.

18. Lane 279: Figure 1 should appear earlier.

19. Lane 282-392: If the authors will not change the title, this part (“4. Oxidative stress as a therapeutic target”) should be deleted and the review and the discussion of the relationship between mitochondrial dysfunction and disease should be strengthened. If the authors will change the title, 

Lane 303-316 are unnecessary.
Lane 317-327 are unnecessary.
Lane 328-341 might be unnecessary.

-The title was changed. The lines were left but rewritten, other Reviewers found them important.

20. Lane 379-392: The reviewer cannot understand the intent of the discussion and ask the authors to rewrite it. Citing the 27 references (123-140) at one time is highly unusual and no use for review.

-This section was shortened and rewritten. Many of the cited  article have been deleted.

21. Figure 2 is not necessary for the reader.

-Because other reviewers think otherwise, we left it in but with a more extensive caption.

22. About “Zilavir”: If the authors have received funding or budget from the manufacturer of the drug or other related sources in the past, the authors should describe it in the Conflicts of Interest section. In the absence of such a past, the reviewer believes that specific product names should be avoided in the text of academic papers and the chemical names of the product should be described.

- We did not receive any sources of financing from the manufacturer of the probiotic, or the properly described dietary supplement, therefore there is no conflict of interest. We pay for open access from the statutory funds of Institutes or research grants. People from the company who participate in writing a job, just like us, have academic titles and we do research together, and what makes us different is the affiliation of the workplace, while the research basis is the same. I agree that specific names of products should be avoided, but apart from the trade name, this is also a common name describing the group of ingredients used, therefore we decided to use this name in our manuscript, but thank you very much for a very valuable and relevant suggestion.

23. Lane 396-404: The reviewer believes that it is inappropriate to cite references in the Conclusion part. The reference 121 and 122 are not best ones and could be replace with more suitable ones.

-They have been deleted.

24. In “Author Contribution”: The reviewer does not understand why there are contributions in methodology, software validation, and resources in a review article. KK (Karol Kramkowski) and KW (Karol Wrzosek) have contributed only in Supervision. Joanna Mikoda (JM) has no contribution in this manuscript and can not be an author of this manuscript. The statement by the authors that all 10 (11-JM) of the authors were in charge of Supervision is unacceptable as a fact, and the reviewer have to distrust the statement by the authors.

-Thank you very much for your suggestions. This is probably a system error that is getting more and more unfortunately. Both gentlemen actively participated not only in publication supervision and consultations, but also participated in the described processes or mechanisms of action. Ms Joanna Mikoda has now been included in the acknowledgments for reliable substantive support during the preparation of the manuscript. And we would like to ask, as all the co-authors of the work, to have greater trust in us as authors, because we devote a lot of work to the verification of the information and issues quoted, and we often consult with all co-authors about the mechanisms and processes cited in order to fully present a given fragment of knowledge to a wide group of readers including not only specialists but also laymen.

Reviewer 2 Report

The authors reviewed mitochondrial respiration, mtDNA/nDNA, and their link to oxidative stress. They further discussed the association of mitochondrial oxidative stress to various diseases. Finally, the authors discussed antioxidants as potential therapeutic strategies. The figures complement the text; however, in terms of the figure legends, the authors should provide sufficient details so that the figures can stand alone. Some information presented in the figures did not mention in the text. The cancer section is weak. The role of the redox system in cancer and neurodegenerative diseases may be different.

Other comments:

  1. Glutathione is a major determinant of cellular redox and thus needs to be emphasized.
  2. Lines 328-332, the example of SOD1 is not appropriate as SOD1 mutation is responsible for the occurrence of ALS, which further confirms the importance of wild-type SOD1. Transgenic mice carrying mutant SOD1-G93A developed ALS with mitochondrial dysfunction and oxidative stress in the brain and the spinal cord. Further studies showed that the mutant SOD1 protein was carbonylated and aggregated.
  3. Line 339-341. Overexpression of GPX1 with resistance to therapy in cancer is not controversial. Increased levels of antioxidant enzymes and transcription factors are often observed in cancer.

Author Response

Reviewer 2

Thank you very much for the apt suggestions that contributed to the improvement of the quality of the manuscript. All comments and suggestions have been taken into account and marked in red or a block in green.

The authors reviewed mitochondrial respiration, mtDNA/nDNA, and their link to oxidative stress. They further discussed the association of mitochondrial oxidative stress to various diseases. Finally, the authors discussed antioxidants as potential therapeutic strategies. The figures complement the text; however, in terms of the figure legends, the authors should provide sufficient details so that the figures can stand alone. Some information presented in the figures did not mention in the text.

-Information not included in the text have been deleted. Legends to figures have been expanded.

The cancer section is weak. The role of the redox system in cancer and neurodegenerative diseases may be different.

-This section has been expanded

Other comments:

1. Glutathione is a major determinant of cellular redox and thus needs to be emphasized.

  • It has been done.

2. Lines 328-332, the example of SOD1 is not appropriate as SOD1 mutation is responsible for the occurrence of ALS, which further confirms the importance of wild-type SOD1. Transgenic mice carrying mutant SOD1-G93A developed ALS with mitochondrial dysfunction and oxidative stress in the brain and the spinal cord. Further studies showed that the mutant SOD1 protein was carbonylated and aggregated.

  • We developed discussion on the possible role of SOD1.

3. Line 339-341. Overexpression of GPX1 with resistance to therapy in cancer is not controversial. Increased levels of antioxidant enzymes and transcription factors are often observed in cancer.

  • We agree with the Reviewer and this has been rewritten.

Reviewer 3 Report

The manuscript of “Mitochondrial Oxidative Stres – A Causative Factor in Many Diseases” by PaweÅ‚ Kowalczyk and co-authors is a well-written and detailed overview of the most important findings that implicated mitochondrial oxidative stress in a number of of pathologies, as well as current advances in antioxidant therapy. The authors focused on the rationale for the use of antioxidants for the treatment of neurodegenerative diseases and cancer. The topic of the review is highly relevant. The manuscript is very interesting and easy to read. The review covers a large amount of literature data and makes a significant contribution to the systematization of knowledge about mitochondrial oxidative stress. The authors cited a large number of recent research articles.

Comments.

1. The authors should describe in more detail the following important issues: 1) the source of excessive formation of ROS in the respiratory chain; 2) the role of reverse electron transport (RET) in the excessive formation of ROS, which can be related to two states of complex I.

2. Fig. 1 shows the diseases that are not mentioned in the main text. It is necessary to fix this. The authors should consider including in Fig. 1 relevant references to the mentioned diseases and / or to the main mechanisms of mitochondrial dysfunction caused by oxidative stress.

3. The headings of the subclauses 1.1. and 1.2 contain the redundant word "mitochondria" that needs to be removed.

4. Line 76: “The mitochondrial crest” needs to be replaced with mitochondrial cristae.

5. Four complexes of the respiratory chain should be written in the traditional way. For example, instead of "NAD-coenzyme Q oxidoreductase", write mitochondrial NADH: ubiquinone oxidoreductase (complex I).

6. The heading of the clause 2: add “-s” to “Mutation”. Three more reasons should be added for the increased vulnerability to mutations of mtDNA compared to nDNA. These reasons are the absence of introns, increased supercoiling, and the attachment of mtDNA nucleoids to the inner side of the inner mitochondrial membrane, that is, directly to the main place of ROS formation.

7. Line 138: The phrase of “mutations in nDNA that codes mitochondrial genes” should be corrected.

8. Lines 303-305: The appropriate references need to be added.

9. Lines 319-320: “The hydrogen pexidase” should be replaced with hydrogen peroxide.

10. Lines 402-404 The phrase should be corrected (… “they increase binding activity of the active form” ….).

Author Response

Reviewer 3

Thank you very much for the apt suggestions that contributed to the improvement of the quality of the manuscript. All comments and suggestions have been taken into account and marked in red or a block in green.

The manuscript of “Mitochondrial Oxidative Stress – A Causative Factor in Many Diseases” by PaweÅ‚ Kowalczyk and co-authors is a well-written and detailed overview of the most important findings that implicated mitochondrial oxidative stress in a number of pathologies, as well as current advances in antioxidant therapy. The authors focused on the rationale for the use of antioxidants for the treatment of neurodegenerative diseases and cancer. The topic of the review is highly relevant. The manuscript is very interesting and easy to read. The review covers a large amount of literature data and makes a significant contribution to the systematization of knowledge about mitochondrial oxidative stress. The authors cited a large number of recent research articles.

Comments.

1.The authors should describe in more detail the following important issues: 1) the source of excessive formation of ROS in the respiratory chain; 2) the role of reverse electron transport (RET) in the excessive formation of ROS, which can be related to two states of complex I.

 -This has been added to section 1.1. Mitochondrial oxidative phosphorylation (OXPHOS) - a source of free radical formation.

2. Fig. 1 shows the diseases that are not mentioned in the main text. It is necessary to fix this. The authors should consider including in Fig. 1 relevant references to the mentioned diseases and / or to the main mechanisms of mitochondrial dysfunction caused by oxidative stress.

-The diseases not mentioned in the text has been deleted, References has been added

3. The headings of the subclauses 1.1. and 1.2 contain the redundant word "mitochondria" that needs to be removed.

-This has been removed.

4. Line 76: “The mitochondrial crest” needs to be replaced with mitochondrial cristae.

-This has been replaced.

5. Four complexes of the respiratory chain should be written in the traditional way. For example, instead of "NAD-coenzyme Q oxidoreductase", write mitochondrial NADH: ubiquinone oxidoreductase (complex I).

-This has been changed according to your suggestion.

6. The heading of the clause 2: add “-s” to “Mutation”. Three more reasons should be added for the increased vulnerability to mutations of mtDNA compared to nDNA. These reasons are the absence of introns, increased supercoiling, and the attachment of mtDNA nucleoids to the inner side of the inner mitochondrial membrane, that is, directly to the main place of ROS formation.

-This has been added to the text.

7. Line 138: The phrase of “mutations in nDNA that codes mitochondrial genes” should be corrected.

- This has been corrected to “.. that codes mitochondrial OXPHOS subunits”.

8. Lines 303-305: The appropriate references need to be added.

-Reference has been added.

9. Lines 319-320: “The hydrogen pexidase” should be replaced with hydrogen peroxide.

 -This has been corrected

10. Lines 402-404 The phrase should be corrected (… “they increase binding activity of the active form” ….).

-This has been corrected

Round 2

Reviewer 1 Report

The manuscript has been considerably improved. However, the manuscript is still immature and some major revisions are essential to be published in IJMS.

Major:
1.  Lane 234, The manuscript lacks description of the relationship between schizophrenia and mitochondrial oxidative stress.

2.  Lane 240, Description in “3.2. Neurodevelopmental Disorders” is weak.

3. Figure 2 is not necessary for this manuscript. There is no description of the environmental mutagens/carcinogens and the structural patterns of the modified bases presented in Figure 2 in the manuscript. Figure 2 is not conducive to the reader's understanding.

4.  Conclusion part should be rewritten, since the current Conclusion is far from conclusive. The reviewer dose not understand what the authors want to say in this review.

5. 
Lane 478, [123–140], 
Lane 485, [16–30,123–140],
Lane 501, [1-171],
Citation of references have to be improved. Basically, the authors should cite the minimum number of references necessary one by one, to let the readers address the references they need. At the same time, the authors should do so, to avoid raising suspicion that the authors are citing their own literatures unnecessarily.
The format of cited references in the reference list should be aligned before publication. The order of the newly added references should be corrected before publication.

Minor: 
Lane 256, “pathomechanism”. 
Lane 435, H2O2-induced. 
The manuscript should be carefully checked for misspelling and grammatical errors before publication.

Author Response

Dear Reviewer

Thank very much for your remarks, that have contributed to improving our work. Please below find our response to you remarks.

The manuscript has been considerably improved. However, the manuscript is still immature and some major revisions are essential to be published in IJMS.

Major:
1.  Lane 234, The manuscript lacks description of the relationship between schizophrenia and mitochondrial oxidative stress.

- This has been completed (text in red color).

  1. Lane 240, Description in “3.2. Neurodevelopmental Disorders” is weak.

- New information has been added (text in red color).

  1. Figure 2 is not necessary for this manuscript. There is no description of the environmental mutagens/carcinogens and the structural patterns of the modified bases presented in Figure 2 in the manuscript. Figure 2 is not conducive to the reader's understanding.

- The reviewer is right, however both other reviewers found them helpful, therefore we prefer to leave them.

  1. Conclusion part should be rewritten, since the current Conclusion is far from conclusive. The reviewer dose not understand what the authors want to say in this review.

-This has been rewritten. We hope that now conclusion is more “conclusive”.

  1. Lane 478, [123–140], 
    Lane 485, [16–30,123–140],
    Lane 501, [1-171],
    Citation of references have to be improved. Basically, the authors should cite the minimum number of references necessary one by one, to let the readers address the references they need. At the same time, the authors should do so, to avoid raising suspicion that the authors are citing their own literatures unnecessarily.The format of cited references in the reference list should be aligned before publication. The order of the newly added references should be corrected before publication.

- Some references have been deleted. References in the text and list have been corrected according to IJMS requirements.

Minor: 
Lane 256, “pathomechanism”. 

-It has been corrected

Lane 435, H2O2-induced. 

-It has been corrected

The manuscript should be carefully checked for misspelling and grammatical errors before publication.

-It has been checked and corrected

Round 3

Reviewer 1 Report

Accepted.

Author Response

Thank you very much for many for many valuable suggestions that improved the quality of our manuscript.
